# Non-Invasive *Mycobacterium avium* Detection Using ^99m^Tc-GSA on Single-Photon Emission Computed Tomography

**DOI:** 10.3390/pharmaceutics17030362

**Published:** 2025-03-13

**Authors:** Yuri Nishiyama, Asuka Mizutani, Masato Kobayashi, Miyu Kitagawa, Yuka Muranaka, Kakeru Sato, Hideki Maki, Keiichi Kawai

**Affiliations:** 1Division of Health Sciences, Graduate School of Medical Sciences, Kanazawa University, 5-11-80 Kodatsuno, Kanazawa 920-0942, Japan; yuri.nishiyama@shionogi.co.jp (Y.N.); m1822kitagawa@stu.kanazawa-u.ac.jp (M.K.); stkk0323@g.u-fukui.ac.jp (K.S.); 2Laboratory for Drug Discovery & Disease Research, Shionogi & Co., Ltd., 3-1-1 Futaba-cho, Toyonaka 561-0825, Japan; hideki.maki@shionogi.co.jp; 3Faculty of Health Sciences, Institute of Medical, Pharmaceutical and Health Sciences, Kanazawa University, 5-11-80 Kodatsuno, Kanazawa 920-0942, Japan; mizutani.a@staff.kanazawa-u.ac.jp (A.M.); kobayasi@mhs.mp.kanazawa-u.ac.jp (M.K.); 4Department of Radiological Technology, Faculty of Health Science, Juntendo University, 2-1-1 Hongo, Bunkyo-ku, Tokyo 113-8421, Japan; y.muranaka.vt@juntendo.ac.jp; 5Radiological Center, University of Fukui Hospital, 23-3 Matsuokashimoaizuki, Eiheiji-cho, Yoshida-gun, Fukui 910-1193, Japan; 6Biomedical Imaging Research Center, University of Fukui, 23-3 Matsuoka-Shimoaizuki, Eiheiji-cho, Yoshida-gun, Fukui 910-1193, Japan

**Keywords:** SPECT, ^99m^Tc-GSA, NTM-PD, MAC, *M. avium*, non-invasive

## Abstract

**Background**: The prevalence of nontuberculous mycobacteria (NTM) infection is on the rise, surpassing that of pulmonary tuberculosis in Japan. Current standard therapy for NTM infection involves long-term treatment of at least 1.5 years, with low success rates and a high relapse rate. ^99m^Tc-diethylenetriaminepentaacetic acid-galactosyl-human serum albumin (^99m^Tc-GSA) is used for human liver imaging. In this study, we utilized ^99m^Tc-GSA as a probe to detect *Mycobacterium avium* (*M. avium*), a major pathogen in NTM pulmonary diseases (NTM-PDs). Our aim was to investigate the non-invasive detection of *M. avium* using ^99m^Tc-GSA on Single-Photon Emission Computed Tomography (SPECT). **Methods**: The accumulation of ^99m^Tc-GSA in *M. avium* was investigated in vitro. In vivo, SPECT images were obtained after the administration of ^99m^Tc-GSA to an *M. avium* thigh infection model. Subsequently, the contrast difference in accumulated ^99m^Tc-GSA between infected and non-infected thighs was calculated using SPECT imaging. Furthermore, SPECT images were obtained for thighs infected with varying bacterial loads, and the accumulation was compared between them. **Results**: In vitro, we observed that ^99m^Tc-GSA accumulates in *M. avium*. In vivo, SPECT images demonstrated the specific accumulation of ^99m^Tc-GSA at the infection site, with this accumulation being correlated with the bacterial load. **Conclusions**: ^99m^Tc-GSA specifically accumulates in *M. avium*, and SPECT can be used to monitor the distribution and quantity of *M. avium* in animals. By utilizing these measures, ^99m^Tc-GSA can be targeted to the site of infection and used as a bacterial probe.

## 1. Introduction

The number of people infected with *Mycobacterium tuberculosis* (*M. tuberculosis*), one of the three major infectious diseases, reached 10.6 million in 2022, exceeding the number in the previous year [1]. Tuberculosis (TB) remains the second leading cause of death as a single infectious disease, with an estimated 1.3 million deaths in 2022 [2]. Nontuberculous mycobacteria (NTM) are mycobacterial species other than *M. tuberculous* and *M. leprae*. In 2014, a survey in Japan reported that the incidence of NTM pulmonary diseases (NTM-PDs) surpassed that of pulmonary TB and had increased rapidly 2.6 times that in the 2007 survey [3]. The number of NTM-PD cases is also increasing worldwide [4,5].

*M. avium* and *M. intracellulare* are collectively termed *M. avium* complex (MAC), and pulmonary MAC infections account for 80–90% of NTM-PD cases in Japan [2]. The standard treatment for pulmonary MAC infections is the triple therapy of clarithromycin or azithromycin, ethambutol, and rifampicin for at least 6–12 months [6,7,8]. However, 48% of patients deemed negative for MAC undergo relapse [9]. Thus, even though treatment for pulmonary MAC infections is long term, the relapse rate is high, and thus, follow-up after treatment is important. Culture conversion is diagnosed if three consecutive sputum cultures are negative, but there are many cases where MAC cannot be cultured from sputum because the condition of the sputum is poor. Even if a MAC colony can be detected, it takes up to six weeks to culture it from sputum [10].

In addition to sputum culture, imaging diagnostics such as computed tomography (CT) and X-rays are used to form a diagnosis and evaluate the effectiveness of treatment for pulmonary MAC disease [6,7,8]. However, these imaging diagnostics only observe the state of the host and cannot evaluate the number or distribution of MAC. Therefore, we investigated the ability to detect *M. avium* non-invasively using Single-Photon Emission Computed Tomography (SPECT), an imaging method used primarily in fields other than infectious diseases [11,12]. If MAC can be detected non-invasively in real time, the presence or absence of MAC in a host can be confirmed even if sputum cannot be collected or MAC cannot be cultured from sputum in clinical settings. The use of SPECT might also reduce the time required for sputum culture, allowing for the early selection of appropriate treatments and drugs, which is expected to improve the prognosis. In non-clinical research, non-invasively detecting MAC on SPECT can be used to monitor the same individual in real-time, making it possible to reduce the number of animals used and shorten evaluation times.

To find the SPECT probe that can be used to detect MAC, we screened the accumulation of 17 SPECT probes in *M. avium* and confirmed that some SPECT probes showed high accumulation. In this study, we selected ^99m^Tc-diethylenetriaminepentaacetic acid-galactosyl-human serum albumin (^99m^Tc-GSA), whose accumulation mechanism in humans was clarified in SPECT probe interview form. ^99m^Tc-GSA is a synthetic glycoprotein where galactose is bound to albumin and is used for the diagnosis of human liver disease.

The binding of ^99m^Tc-GSA to *M. avium* was evaluated in vitro. In vivo, SPECT images were obtained after the administration of ^99m^Tc-GSA to an *M. avium* thigh infection model. Using the SPECT image, the contrast difference in the accumulation of ^99m^Tc-GSA between the infected and non-infected sites was calculated. SPECT images were obtained with varying numbers of infected bacteria, and the accumulation was compared between sites.

## 2. Materials and Methods

### 2.1. Microorganisms

*M. avium* ATCC700898 was cultured on Middlebrook 7H9 Broth (Becton, Dickinson and Co., Franklin Lakes, NJ, USA) and incubated at 37 °C for 4 d with shaking. The number of colonies was counted based on optical density (OD) measurements and used for an in vitro accumulation assay. The number of viable cells was determined through plating on Middlebrook 7H10 Agar (Becton, Dickinson and Co.). A stock solution that was stored at −80 °C was used for the animal tests.

### 2.2. Animals

Specific-pathogen-free male ICR mice (CLEA Japan Inc., Tokyo, Japan, 5 weeks old) were used for all in vivo studies. The Institutional Animal Care and Use Committee of Shionogi & Co., Ltd. (Osaka, Japan), approved all animal study procedures.

### 2.3. In Vitro Accumulation of ^99m^Tc-GSA in M. avium ATCC700898

*M. avium* ATCC700898 was precultured on Middlebrook 7H9 Broth for 4 d. Thereafter, the bacterial solution was adjusted to OD = 0.10 with phosphate-buffered saline (PBS; pH 7.4; Takara Bio, Shiga, Japan). Then, 10 µL of 74 kBq ^99m^Tc-GSA (Nihon Mediphysics, Tokyo, Japan) was added to *M. avium* ATCC700898 solution and incubated at 37 °C for 5, 15, 30, 60, 120, and 240 min. After that, the solution was centrifuged at 4 °C at 7000× *g* for 10 min, and the supernatant was removed. PBS was added to it, and it was washed twice using the same method as above. After washing, 1.0 mL of 0.1 N NaOH aqueous solution (Nacalai Tesque, Kyoto, Japan) was added and mixed to lyse the bacteria. Then, the radioactivity of *M. avium* ATCC700898 was measured using a gamma counter (WIZARD2480, PerkinElmer, Waltham, MA, USA). The following formula was used to calculate the accumulation rate: accumulation rate (%ID) = counts of sample (cpm)/counts of injected radioactivity (cpm) × 100.

### 2.4. Effect of the Asialoglycoprotein Receptor (ASGP-R) Inhibitor on ^99m^Tc-GSA Accumulation

ASGP-R, which is only expressed by mammalian hepatocytes, recognizes the galactose residue of ASGP and imports ASGP into hepatocytes [13]. In humans, it has been shown that ^99m^Tc-GSA, as with ASGP, is imported by ASGP-R [14]. Therefore, we investigated the accumulation of ^99m^Tc-GSA by adding asialofetuin, an inhibitor of ASGP-R [15]. Asialofetuin (Sigma-Aldrich, St. Louis, MO, USA) was dissolved with distilled water to a final concentration of 0.01, 0.1, or 1 mg/mL. Prepared asialofetuin and 74 kBq/10 µL of ^99m^Tc-GSA were added to *M. avium* ATCC700898 at the same time and incubated for 1 h at 37 °C. After that, radioactivity was measured as above and the accumulation rate was calculated.

### 2.5. Biodistribution of ^99m^Tc-GSA

First, 0.2 mL of 10 kBq ^99m^Tc-GSA was injected into the mice via the tail vein. At 1, 2, and 4 h post-injection, their blood was collected under anesthesia and their organs were removed. The organ weight was measured, and the radioactivity of the blood and organs was measured. The accumulation of each organ is shown as a percentage of the ID per gram of wet tissue (%ID/g, Table 1).

### 2.6. SPECT Imaging of ^99m^Tc-GSA in an M. avium ATCC700898 Mouse Thigh Infection Model

First, 0.1 mL of 1.0 × 10^8^ cfu *M. avium* ATCC700898 was injected into the left thigh. At 1 h after infection, 0.2 mL of 10–20 MBq ^99m^Tc-GSA was injected. ^99m^Tc-GSA accumulation at the infection site was imaged using SPECT/CT (Triumph II SPECT 2H/XO SRI CT, TriFoil Imaging, Los Angeles, CA, USA). At 1, 2, and 4 h after ^99m^Tc-GSA injection, the mice were arranged on a SPECT/CT bed and fixed with surgical tape under anesthesia with isoflurane. SPECT imaging was acquired under the same conditions as in [16]. Then, the thigh was removed and taken for SPECT imaging in the same way. Regions of interest were drawn over each thigh and for image processing. After attenuation correction, the %ID was calculated using both the radioactivity on each thigh and the administered radioactivity. The contrast was calculated by dividing the accumulation in the left thigh by the accumulation in the right thigh.

### 2.7. Counting of Viable Bacteria

The entire infected thigh muscle was removed and homogenized. Serial dilutions of the samples were then plated on 7H10 Agar with 5% Middlebrook OADC Enrichment (Becton, Dickinson and Co.). The plates were then incubated at 35 °C for 3–4 weeks. Then, the number of colonies was counted and the number of viable cells in the thigh (log10 CFU/thigh) was calculated [17].

### 2.8. Statistical Analysis

The difference in accumulation was evaluated using the Student’s *t*-test. *p* < 0.05 was considered statistically significant and accepted within 95% confidence limits using SAS^®^ Studio (SAS Institute Inc., Cary, NC, USA). All results are reported as means ± standard deviation (SD).

## 3. Results

### 3.1. In Vitro Accumulation of ^99m^Tc-GSA in M. avium ATCC70089

Figure 1 shows the ^99m^Tc-GSA accumulation in *M. avium* ATCC700898 incubated for different durations. At 5, 15, 30, 60, 120, and 240 min, ^99m^Tc-GSA accumulation was 1.38%, 1.88%, 1.99%, 3.61%, 4.81%, and 4.95% of the ID, respectively. Thus, the accumulation of ^99m^Tc-GSA increased as the incubation time increased.

### 3.2. Effect of the Asialoglycoprotein Receptor (ASGP-R) Inhibitor on ^99m^Tc-GSA Accumulation

We examined the effect of adding the ASGP-R inhibitor asialofetuin on ^99m^Tc-GSA accumulation. Figure 2 shows the accumulation rate upon the addition of various concentrations of asialofetuin. ^99m^Tc-GSA accumulation was reduced in correlation with the inhibitor concentration.

### 3.3. Biodistribution of ^99m^Tc-GSA

Biodistribution analysis showed that the blood concentration of ^99m^Tc-GSA was approximately 0.5% 1 h after administration, and the blood clearance was fast. In terms of its distribution to the organs, the accumulation of ^99m^Tc-GSA was highest in the liver, at 11–12% at all evaluation times, followed by the kidney, albeit at a lower level (Table 1).

### 3.4. SPECT Imaging of ^99m^Tc-GSA in an M. avium ATCC700898 Mouse Thigh Infection Model

Figure 3 shows a representative SPECT image of an *M. avium* ATCC700898 thigh infection model mouse. The left thigh, indicated by the red arrow, is the infection site. The area with a high accumulation of ^99m^Tc-GSA at the top of the image is thought to be the intestine, which is the main excretion site of ^99m^Tc-GSA [18]. As can be seen from the figure, ^99m^Tc-GSA clearly accumulated at the infection site (left thigh, red arrow), compared with the non-infected site (right thigh).

Figure 4 shows SPECT images of the thigh over time following ^99m^Tc-GSA administration. As with the whole-body images, the infection site (left thigh) showed greater ^99m^Tc-GSA accumulation compared with the non-infected site (right thigh) at all time points after ^99m^Tc-GSA administration. Of note, the shorter the time since ^99m^Tc-GSA administration, the greater the ^99m^Tc-GSA accumulation at the infection site.

Table 2 shows the accumulation of ^99m^Tc-GSA in each thigh measured based on ROI in SPECT images and the contrast values between sites [16]. The accumulation rates at the infection site were 1.73, 1.22, and 0.88%ID at 1, 2, and 4 h after infection, respectively. The accumulation rates at the non-infected sites were 0.33, 0.31, and 0.29%ID, respectively. The percentage of accumulation at both sites decreased with time after ^99m^Tc-GSA administration. The contrast of accumulation at the infected and non-infected sites was 5.40, 3.96, and 3.07 at 1, 2, and 4 h after infection, respectively. The contrast decreased with time after ^99m^Tc-GSA administration.

Finally, the contrast in accumulation from the SPECT images (Figure 5) was evaluated when the bacterial load was varied (Table 3). Accumulation at the infection site was 1.73 and 1.13% ID/g at 10^8^ and 10^5^ CFU/mouse. Accumulation at non-infected sites was 0.33 and 0.36% ID/g, and the accumulation at both sites decreased with time. The contrast between the infected and non-infected sites was 5.40 and 3.11 at 10^8^ and 10^5^ CFU/mouse, and the contrast decreased in correlation with the number of bacteria used for infection.

## 4. Discussion

In this study, we investigated the ability to detect *M. avium* non-invasively using ^99m^Tc-GSA on SPECT. Although triple-drug combination therapy has been established as a standard treatment for pulmonary NTM infection, the disease requires a long treatment period of at least 18 months. In addition, the therapeutic effect of the standard therapy is not sufficient, with a reported success rate of only 60% [19]. Therefore, the development of a new, safe, and effective antibiotic is desired. When conducting in vivo research to discover new antibacterial drugs in a non-clinical setting, it is common to measure the number of bacteria at the infection site of an animal to confirm whether the compound has the effect of reducing the number of bacteria [20]. However, this method requires animals to be dissected over time at each evaluation point, which requires a large number of animals. In addition, in infectious diseases with a long infection period, such as pulmonary MAC disease, the bacteria may form a biofilm [21], and the infection state may differ between individuals, associated with differences in the immune state of the host. Therefore, the accurate evaluation of infection that considers individual differences in long-term infection models or chronic infection models can be performed by evaluating the same individual over time.

As shown in Figure 1, the accumulation of ^99m^Tc-GSA in *M. avium* and the accumulation rate increased with increasing incubation time. The doubling time of *M. avium* was longer than the incubation time in this study [22], and thus, it is unlikely that the number of bacteria increased. Therefore, the increase in accumulation was not related to an increase in the number of bacteria, but rather that it took time for the bacteria to accumulate. Figure 2 shows that the accumulation of ^99m^Tc-GSA decreased as the concentration of asialofetuin, an ASGP-R inhibitor, increased. This suggests that ^99m^Tc-GSA competitively inhibits asialofetuin. The results of this study suggest that ^99m^Tc-GSA accumulates in *M. avium* via an asialofetuin-sensitive binding site on the cell membrane of *M. avium*, and competitive inhibition occurs at this binding site. Further investigation is required to clarify the mechanism involved in the accumulation of ^99m^Tc-GSA in *M. avium*. Biodistribution analysis showed high accumulation of ^99m^Tc-GSA in the liver, probably because ASGP-R is also present in the mouse liver [23]. ^99m^Tc-GSA accumulated in the kidney at a lower level compared with that in the liver.

The accumulation analyses showed that in the *M. avium* thigh infection model, ^99m^Tc-GSA accumulated more at the infected site than at the non-infected site, and the contrast was also high, suggesting that ^99m^Tc-GSA accumulates specifically at the infection site. In addition, the contrast was higher when the time from ^99m^Tc-GSA administration was shorter. One of the reasons for this is that if we assume that ^99m^Tc-GSA accumulates in *M. avium* by binding to its surface, accumulation at the infection site would decrease as the blood concentration decreases because surface binding is an equilibrium reaction. Furthermore, the accumulation of ^99m^Tc-GSA increased with the greater number of inoculated bacteria. This suggests the possibility of non-invasively monitoring the distribution and number of bacteria in a host, as demonstrated in the *M. avium* infection model in this study. SPECT enables the non-invasive monitoring of the same individual, which is important when studying chronic infection models where there is considerable inter-individual variability, and thus, it is likely to be a useful tool when developing drugs to treat chronic infections.

These results demonstrate that ^99m^Tc-GSA specifically accumulates in bacteria and that ^99m^Tc-GSA can be delivered specifically to the site of infection as a bacterial probe.

In clinical practice, a certain number of NTM-PD patients gradually deteriorate after follow-up and begin treatment [24]. Considering the high sensitivity of nuclear medicine imaging, including SPECT, non-invasive monitoring of bacterial load in these patients would allow for the identification of an increase in bacterial load before CT imaging or symptoms worsen, allowing for more precise timing of treatment initiation. In addition, the rate of MAC patients not responding to standard therapy is approximately 30% [25], and the therapeutic efficacy of adding Alikayce, an inhaled form of AMK, is not satisfactory [26]. By monitoring bacterial load during long-term administration, it is possible to promptly assess the effectiveness of the therapy and select the most appropriate drug at an early stage.

## 5. Conclusions

This study shows that ^99m^Tc-GSA specifically accumulates in *M. avium*, and SPECT can be used to monitor the distribution and quantity of *M. avium* in animals. By utilizing these measures, ^99m^Tc-GSA can be targeted to the site of infection and used as a bacterial probe. Furthermore, by applying this technology to drug discovery research of *M. avium*, which causes a slow progressive disease, it will be possible to monitor the same individual over time, eliminating the need to consider individual differences and enabling research to be conducted with a minimal number of animals. In addition, ^99m^Tc-GSA is currently used in humans, and thus, hurdles to its application in humans are low, and it may be used to diagnose pulmonary MAC infection and evaluate the effectiveness of treatments. Future research should investigate its application to pulmonary infections and the evaluation of other NTM species.

## Figures and Tables

**Figure 1 pharmaceutics-17-00362-f001:**
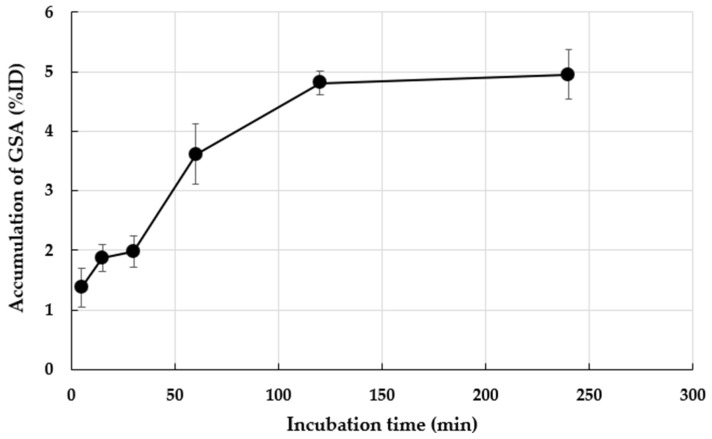
In vitro accumulation of ^99m^Tc-GSA in *M. avium* ATCC700898. Accumulation is expressed as the means ± standard deviation (SD) of three tests.

**Figure 2 pharmaceutics-17-00362-f002:**
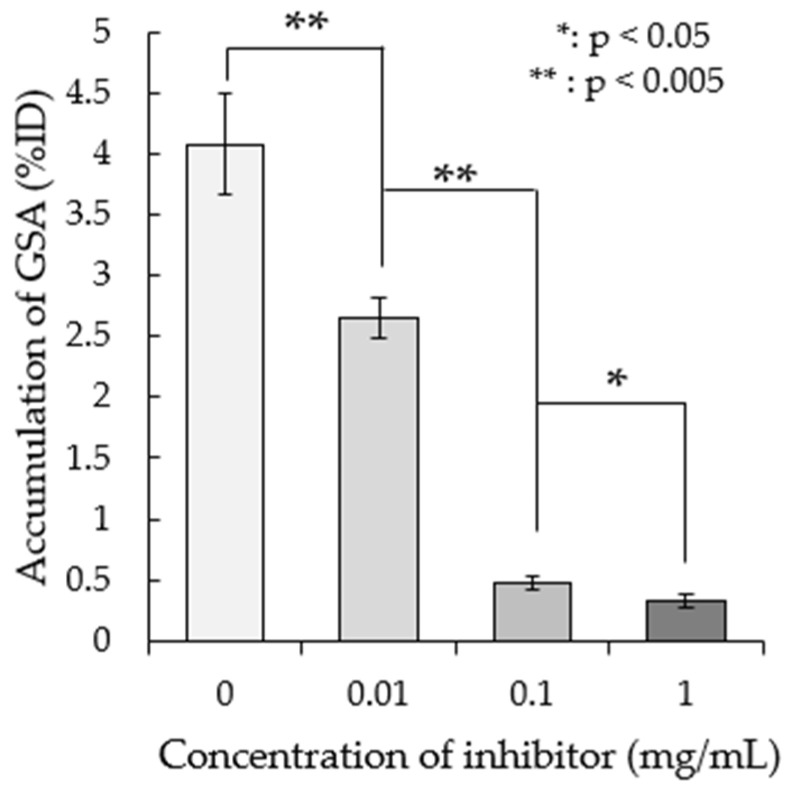
In vitro accumulation ratio change by concentration of the inhibitor. The results are expressed as the means ± SD of three tests.

**Figure 3 pharmaceutics-17-00362-f003:**
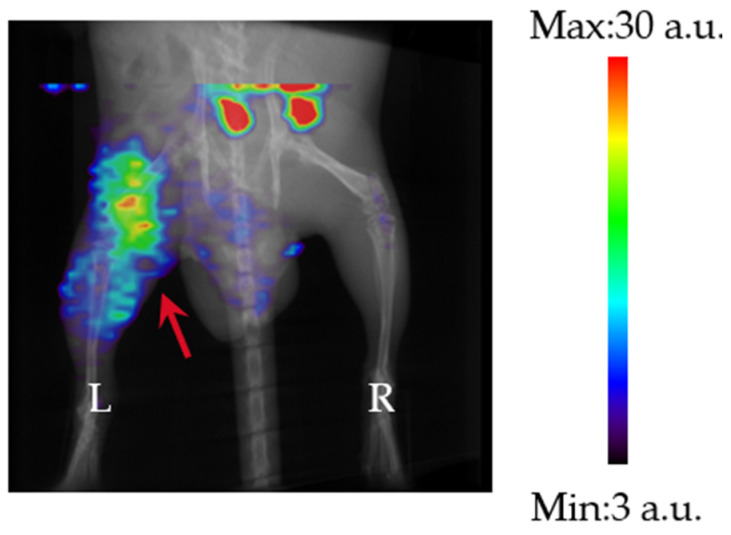
SPECT image (Maximum Intensity Projection (MIP)) of ^99m^Tc-GSA in an *M. avium* ATCC700898 mouse thigh infection model (1 h). The red arrow shows the infection site.

**Figure 4 pharmaceutics-17-00362-f004:**
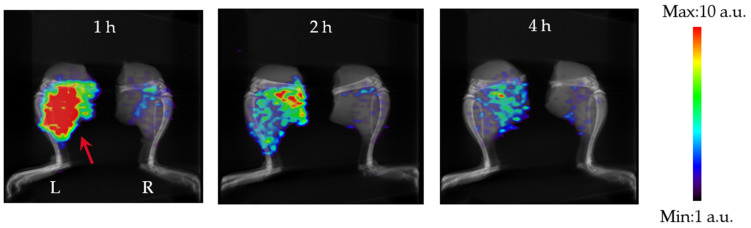
SPECT images (MIP) of the ^99m^Tc-GSA-only thigh by time. The red arrow shows the infection site. These images are shown at the same scale.

**Figure 5 pharmaceutics-17-00362-f005:**
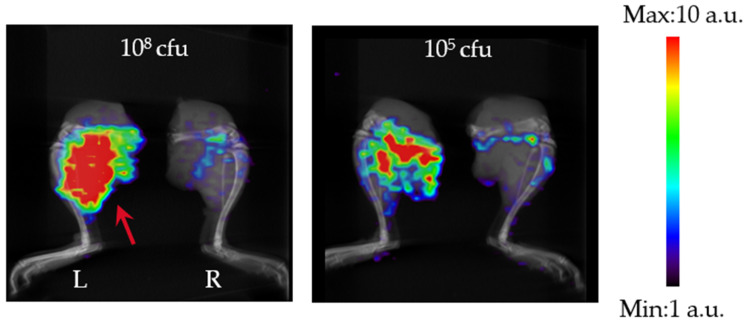
SPECT images (MIP) of the ^99m^Tc-GSA-only thigh at 10^8^ and 10^5^ CFU/mouse (1 h). The red arrow shows the infection site. These images are shown at the same scale.

**Table 1 pharmaceutics-17-00362-t001:** Biological distribution of ^99m^Tc-GSA by time.

	^99m^Tc-GSA Accumulation (%ID/g)		
Organ	1 h	2 h	4 h
Heart	0.35 ± 0.06	0.26 ± 0.06	0.19 ± 0.04
Lung	0.44 ± 0.05	0.31 ± 0.03	0.22 ± 0.04
Liver	11.50 ± 1.10	12.71 ± 2.03	11.14 ± 1.52
Kidney	0.96 ± 0.07	1.16 ± 0.20	1.11 ± 0.20
Blood	0.52 ± 0.19	0.34 ± 0.11	0.32 ± 0.17

Each value is the mean ± SD (*n* = 3).

**Table 2 pharmaceutics-17-00362-t002:** Accumulation and contrast of ^99m^Tc-GSA from SPECT imaging.

**Time After Injection (h)**		**Accumulation (%ID)**	**Contrast**
1	Infected	1.73 ± 0.20	5.40
	Uninfected	0.33 ± 0.07	
2	Infected	1.22 ± 0.21	3.96
	Uninfected	0.31 ± 0.07	
4	Infected	0.88 ± 0.31	3.07
	Uninfected	0.29 ± 0.10	

Expressed as a % of the injected dose. Each value is the mean ± SD (*n* = 3).

**Table 3 pharmaceutics-17-00362-t003:** SPECT analysis of the accumulation and contrast of ^99m^Tc-GSA in the *M. avium* ATCC700898 thigh infection model mice with varying bacterial load in SPECT.

CFU/Mouse		Accumulation (%ID)	Contrast
10^8^	Infected	1.73 ± 0.20 *	5.40
	Uninfected	0.33 ± 0.07	
10^5^	Infected	1.13 ± 0.19	3.11
	Uninfected	0.36 ± 0.06	

Expressed as a % of the injected dose. Each value is the mean ± SD (*n* = 3). * *p* < 0.05 compared with the 10^5^ CFU-infected thigh.

## Data Availability

All data are available in the article.

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
