# Peer review of "Non-Invasive Mycobacterium avium Detection Using 99mTc-GSA on Single-Photon Emission Computed Tomography"

_pharmaceutics, 2025, doi:10.3390/pharmaceutics17030362_

Round 1

Reviewer 1 Report

Comments and Suggestions for Authors

In the current study, a liver imaging developer 99mTc-GSA was used for the detection of Mycobacterium avium. By creating a mouse model of Mycobacterium avium infection and performing SPECT imaging, the researchers confirmed that 99mTc-GSA accumulation was evident at the injection site of pathogenic bacteria in the model mice. Therefore, 99mTc-GSA could be used as a bacterial probe to detect mycobacterium avium infection lesions. This is a very interesting study. However, there are some puzzling problems in the study, and I hope the authors can give further explanation.

  1. Although this study showed a high accumulation of 99mTc-GSA at the site of injection of M. avium, the results of in vitro cell experiments also suggested that the binding of 99mTc-GSA to M. avium may be related to the expression of ASGP-R. However, since its imaging mechanism is not yet clear, it is inevitable to wonder whether 99mTc-GSA can also accumulate in Mycobacterium tuberculosis or other pathogens. Unfortunately, the article does not explain this problem.
  2. The infection model in the study was made by injecting Mycobacterium avium into the thigh muscle of mice and treating it as an infection focus 1 hour later, which is different from the infection process that actually occurs in vivo. The real infection process includes a variety of pathological changes such as inflammatory cell infiltration, edema, cell necrosis, fibrosis, etc., and changes with time. As a specific SPECT imaging agent, its uptake by the target tissue is usually relatively constant during the detectable time, and with the extension of time, the non-specific uptake is gradually cleared, and the lesions are more clearly displayed. However, Figure4 and Table2 in this article show the opposite result. Although the lesion showed clearly at 1 hour after injection, the concentration of 99mTc-GSA gradually decreased with the extension of time. The occurrence of this phenomenon may be related to the absorption of locally injected dense pathogenic bacteria into the blood circulation over time, and may also be related to the stability of 99mTc-GSA binding to Mycobacterium avium. The reasons remain to be further explored.
  3. SPECT imaging results of infected mouse models were presented as local images. Although liver and intestinal uptake may interfere with the observation of thigh infection lesions, whole-body imaging can help to determine the feasibility of 99mTc-GSA as an inflammatory imaging agent.

Author Response

Comments 1: Although this study showed a high accumulation of 99mTc-GSA at the site of injection of M. avium, the results of in vitro cell experiments also suggested that the binding of 99mTc-GSA to M. avium may be related to the expression of ASGP-R. However, since its imaging mechanism is not yet clear, it is inevitable to wonder whether 99mTc-GSA can also accumulate in Mycobacterium tuberculosis or other pathogens. Unfortunately, the article does not explain this problem.

Response 1: Accumulation in the M. intracellurare has been confirmed, but further investigation is required to determine whether it can be applied to other Mycobacteria. On the other hand, accumulation in other bacteria such as S. aureus, P. aeruginosa, and E. coli has been confirmed. We added discussion about it in introduction.

Comments 2: The infection model in the study was made by injecting Mycobacterium avium into the thigh muscle of mice and treating it as an infection focus 1 hour later, which is different from the infection process that actually occurs in vivo. The real infection process includes a variety of pathological changes such as inflammatory cell infiltration, edema, cell necrosis, fibrosis, etc., and changes with time. As a specific SPECT imaging agent, its uptake by the target tissue is usually relatively constant during the detectable time, and with the extension of time, the non-specific uptake is gradually cleared, and the lesions are more clearly displayed. However, Figure4 and Table2 in this article show the opposite result. Although the lesion showed clearly at 1 hour after injection, the concentration of 99mTc-GSA gradually decreased with the extension of time. The occurrence of this phenomenon may be related to the absorption of locally injected dense pathogenic bacteria into the blood circulation over time, and may also be related to the stability of 99mTc-GSA binding to Mycobacterium avium. The reasons remain to be further explored.

Response 2: As you pointed out, the accumulation in the infection site decreased over time. In the in vitro test at Figure 1, the accumulation is thought to have increased because 99mTc-GSA was always present in the reaction system. In the in vivo infection model in Figure 4 and table 2, if 99mTc-GSA accumulates in the MAC by binding to the bacterial surface, the binding to the surface is generally an equilibrium reaction, and it is thought that the accumulation at the infection site decreased as the blood concentration decreased. In addition, regarding the infection model, the primary purpose of this study was to detect M. avium present in the body, therefore we set the time point as 1 hour after infection. As you commented, the actual infection process is more complex, so we need to study chronic infection in the future.

Comments 3: SPECT imaging results of infected mouse models were presented as local images. Although liver and intestinal uptake may interfere with the observation of thigh infection lesions, whole-body imaging can help to determine the feasibility of 99mTc-GSA as an inflammatory imaging agent.

Response 3: 99mTc-GSA physiologically accumulates highly in the liver and intestines, but we believe that it would be less affected if the infection affected the limb muscles, as in this infection model. Furthermore, because mice are small, even if they are affected to some extent, it is possible to take measures such as shielding the trunk with lead plates, and in humans, the trunk and limbs are sufficiently separated, so we believe that the impact would be even smaller.

Reviewer 2 Report

Comments and Suggestions for Authors

General Comments:

Nishiyama et al. present an interesting and innovative approach to the non-invasive detection of Mycobacterium aviumusing 99mTc-GSA and SPECT imaging. Given the rising prevalence of nontuberculous mycobacteria (NTM) infections and the challenges associated with their diagnosis and treatment, this study is timely and potentially impactful. The authors provide a clear rationale for their work and describe both in vitro and in vivo investigations that demonstrate the feasibility of using 99mTc-GSA as a bacterial probe. While the study is well-conceived, some aspects require further clarification and refinement. I recommend minor revisions to improve clarity, strengthen the analysis, and provide additional discussion on the method’s specificity and potential clinical applications.

Specific Comments:

  1. More background on the mechanism by which 99mTc-GSA interacts with avium would be useful.
  2. It would be beneficial to discuss any potential limitations of using the thigh infection model. As lung is the primary infection organ, any plan to test it in pulmonary infection models?
  1. Were any off-target accumulations observed in host tissues? If so, how might this affect specificity?
  2. Was there a statistically significant correlation between bacterial load and 99mTc-GSA accumulation? Provide representative images of SPECT scans for varying bacterial loads to visually support the conclusions if possible.
  3. Discuss whether 99mTc-GSA could be broadly applicable to other Mycobacterium species or if its specificity is limited to M. avium.
  4. In figure 3 and 4, please add units and values to the scale bar.

Author Response

Comments 1: More background on the mechanism by which 99mTc-GSA interacts with avium would be useful.

Response 1: There is currently no additional information about the mechanism and further investigation is required to clarify the mechanism.

Comments 2: It would be beneficial to discuss any potential limitations of using the thigh infection model. As lung is the primary infection organ, any plan to test it in pulmonary infection models?

Response 2: In this report, we investigated thigh infections, which are easy to analyze, but in the future, we plan to investigate lung infections. In lung infection model, it will be difficult to analyze the amount of accumulation due to overlap and accumulation with other organs, so this needs to be taken into consideration.

Comments 3: Were any off-target accumulations observed in host tissues? If so, how might this affect specificity?

Response 3: In the muscle infection model like the thigh infection, we believe that a certain level of image contrast can be obtained regardless of the location. However, as mentioned in comments 2, it is important to note that image contrast is difficult to obtain in organs where 99mTc-GSA physiologically accumulates, such as the liver and kidneys.

Comments 4: Was there a statistically significant correlation between bacterial load and 99mTc-GSA accumulation? Provide representative images of SPECT scans for varying bacterial loads to visually support the conclusions if possible.

Response 4: Thank you for this comment. I analyzed and added data about it.

Comments 5: Discuss whether 99mTc-GSA could be broadly applicable to other Mycobacterium species or if its specificity is limited to M. avium.

Response 5: Accumulation in the M. intracellurare has been confirmed, but further investigation is required to determine whether it can be applied to other Mycobacteria. On the other hand, accumulation in other bacteria such as S. aureus, P. aeruginosa, and E. coli has been confirmed. We added discussion about it in introduction.

Comments 6: 6.   In figure 3 and 4, please add units and values to the scale bar.

Response 5: I optimized the figures.

Reviewer 3 Report

Comments and Suggestions for Authors

The manuscript by Yuri Nishiyama et al. submitted to Pharmaceutics described the

Possibilities of 99mTc-GSA as a probe to detect Mycobacterium avium (M. avium).

 99mTc-GSA can specifically accumulate in M. avium, and SPECT can be utilized to monitor the distribution and quantity of M. avium in mice. The study was well designed, and the experimental results supported the conclusion of the manuscript. Therefore, it is recommended for publication with the following issues being addressed appropriately.

Some revisions are as follows:

  1. How about are the Rf values of [99mTc]TcO4-, [99mTc]TcO2.nH2O, [99mTc]Tc-GSA by TLC?
  2. The authors should explain why they chose 99mTc-GSA as a probe to detect Mycobacterium avium in introduction section. Are there some literatures to support this?
  3. How about is the specific activity of [99mTc]Tc-GSA?

Author Response

Comments 1: How about are the Rf values of [99mTc]TcO4-, [99mTc]TcO2.nH2O, [99mTc]Tc-GSA by TLC?

Response 1: Although we did not check by TLC, we purchase medicines that are used in clinical trials and use them on the day of the test. Therefore, we assume that the radiochemical purity is very high.

Comments 2: The authors should explain why they chose 99mTc-GSA as a probe to detect Mycobacterium avium in introduction section. Are there some literatures to support this?

How about is the specific activity of [99mTc]Tc-GSA?

Response 2: There was no report, but our research group investigated and evaluated the accumulation of 17 SPECT probes in M. avium, confirming that several SPECT probes showed high accumulation. From these, we selected those whose accumulation mechanism in humans had been clarified in the SPECT probe interview form. We added explanation in introduction section.

We purchase medicines that are used in clinical trials and use them on the day of the test, but specific activity values not published.

Reviewer 4 Report

Comments and Suggestions for Authors

Manuscript Number: pharmaceutics-3467689

The manuscript “Non-invasive Mycobacterium avium detection using 99mTc-GSA on SPECT” by Yuri Nishiyama et al. describes the in vivo evaluation of 99mTc-GSA as a probe to detect Mycobacterium avium (M. avium), in experimental animals.

The study is well designed and executed, the manuscript is well organized and deserves publication in Pharmaceutics as an article.

Some minor points:

Lines 111-112 and elsewhere: “The accumulation rate….” Maybe change to “The uptake…”

Line 131: “a percentage of the ID per gram of wet tissue (%ID/organ, Table 1).,

Lines 189-194: Table 1, 99mTc-GSA accumulation (%ID/g) and elsewhere. Please check for consistency ID or ID/g

Line 169: “as the culture time …” maybe change to “as the incubation time …”

Line 189: “that the blood concentration of 99mTc-GSA was approximately 0.5% 1 h after administration, and it disappeared from the blood quickly.” The blood clearance is fast, however it is not correct to say that it disappeared quickly as it drops from 0.5 to 0.3 at 2 and 4hrs post injection, so please rephrase the sentence.

Line 221: “Table 2 shows SPECT images showing…” There are no images in Table 2.

Lines 223-225: The uptake values are given as ID/g. If this is the case then the it should be mention in the experimental that the thighs were weighted. Otherwise use the ID and explain how it is calculated (e.g. ROI, measured in dose calibrator…) 

Author Response

Lines 111-112 and elsewhere: “The accumulation rate….” Maybe change to “The uptake…”

Response: We suggested that 99mTc-GSA accumulates in M. avium via an asialofetuin-sensitive binding site on the cell membrane of the M. avium, but further investigation is required to clarify the mechanism, therefore we choose the term “accumulation rate” rather than “uptake”.

Line 131: “a percentage of the ID per gram of wet tissue (%ID/organ, Table 1).,

Lines 189-194: Table 1, 99mTc-GSA accumulation (%ID/g) and elsewhere. Please check for consistency ID or ID/g

Response: Thank you for the comment. We unified “%ID/g” in biological distribution test. We analyzed “%ID” in in vitro test and SPECT imaging.

Line 169: “as the culture time …” maybe change to “as the incubation time …”

Response: Thank you for the comment. We fixed it.

Line 189: “that the blood concentration of 99mTc-GSA was approximately 0.5% 1 h after administration, and it disappeared from the blood quickly.” The blood clearance is fast, however it is not correct to say that it disappeared quickly as it drops from 0.5 to 0.3 at 2 and 4hrs post injection, so please rephrase the sentence.

Response: Thank you for the comment. We fixed it.

Line 221: “Table 2 shows SPECT images showing…” There are no images in Table 2.

Response: Thank you for the comment. We fixed it.

Lines 223-225: The uptake values are given as ID/g. If this is the case then the it should be mention in the experimental that the thighs were weighted. Otherwise use the ID and explain how it is calculated (e.g. ROI, measured in dose calibrator…)

Response: It is “ID%”, is not “ID/g”. We fixed it.